# Vascular-derived SPARC and SerpinE1 regulate interneuron tangential migration and accelerate functional maturation of human stem cell-derived interneurons

Matthieu Genestine[1], Daisy Ambriz[1], Gregg W Crabtree[2], Patrick Dummer[1], Anna Molotkova[1], Michael Quintero[1], Angeliki Mela[1], Saptarshi Biswas[2], Huijuan Feng[3], Chaolin Zhang[3], Peter Canoll[1], Gunnar Hargus[1], Dritan Agalliu[1,2], Joseph A Gogos[4,5], Edmund Au[1,6]*

[1]Department of Pathology and Cell Biology, Columbia University, New York, United States; [2]Department of Neurology, Columbia University Irving Medical Center, New York, United States; [3]Department of Department of Systems Biology, Columbia University Irving Medical Center, New York, United States; [4]Department of Cellular Physiology and Biophysics, Columbia University, New York, United States; [5]Department of Neuroscience, Zuckerman Mind Brain and Behavior Institute, Columbia University, New York, United States; [6]Columbia Translational Neuroscience Initiative Scholar, New York, United States

*For correspondence:
ea2515@cumc.columbia.edu

Competing interests: The authors declare that no competing interests exist.

**Abstract** Cortical interneurons establish inhibitory microcircuits throughout the neocortex and their dysfunction has been implicated in epilepsy and neuropsychiatric diseases. Developmentally, interneurons migrate from a distal progenitor domain in order to populate the neocortex – a process that occurs at a slower rate in humans than in mice. In this study, we sought to identify factors that regulate the rate of interneuron maturation across the two species. Using embryonic mouse development as a model system, we found that the process of initiating interneuron migration is regulated by blood vessels of the medial ganglionic eminence (MGE), an interneuron progenitor domain. We identified two endothelial cell-derived paracrine factors, SPARC and SerpinE1, that enhance interneuron migration in mouse MGE explants and organotypic cultures. Moreover, pre-treatment of human stem cell-derived interneurons (hSC-interneurons) with SPARC and SerpinE1 prior to transplantation into neonatal mouse cortex enhanced their migration and morphological elaboration in the host cortex. Further, SPARC and SerpinE1-treated hSC-interneurons also exhibited more mature electrophysiological characteristics compared to controls. Overall, our studies suggest a critical role for CNS vasculature in regulating interneuron developmental maturation in both mice and humans.

## Introduction

Cortical interneurons are inhibitory, locally projecting cells that form a distributed network of repetitive circuits throughout the cortex (*Kepecs and Fishell, 2014*; *Tremblay et al., 2016*). To establish this distributed network, interneurons migrate from the ventral subpallium to arealize throughout the cortex in order to form connections with layered pyramidal neurons and other interneurons. The timescale for interneuron migration varies widely across species. The process lasts days in mice (*Lavdas et al., 1999*; *Marín and Rubenstein, 2001*) and several months in humans (*Arshad et al., 2016*; *Hansen et al., 2013*; *Ma et al., 2013*). This species difference extends to stem cell-derived interneurons where mouse ES-derived interneurons develop rapidly (*Au et al., 2013*; *Maroof et al.,*

*2010*; *McKenzie et al., 2019*), while human stem cell-derived interneuron (hSC-interneuron) development is protracted (*Nicholas et al., 2013*; *Shao et al., 2019*). Indeed, this limitation has restricted the scope of analysis on hSC-interneurons, hampering detailed functional studies.

In this study, we sought to identify factors that regulate the timing of interneuron migration across species. Using embryonic mouse development as a model system, we found that interneuron migration coincides with vascularization of the medial ganglionic eminence (MGE), and that by manipulating the degree of MGE vascularization in vivo, it regulates the degree of interneuron migration into the cortex. Using an in vitro approach, we identified paracrine factors, SPARC and SerpinE1, produced by endothelial cells that enhance interneuron migration in mouse MGE explants and organotypic cultures. Given the slow developmental rate of hSC-interneurons, we tested whether SPARC and SerpinE1 treatment similarly induced migration in human cells. We found that treated hSC-interneurons xenografted into neonatal mouse cortex exhibited enhanced migration. Further, transplanted hSC-interneurons also showed greater morphological complexity, and more mature electrophysiological characteristics compared to controls. These data suggest that interneurons in mice and humans share a common vascular-based mechanism that regulates their developmental timing. And, by characterizing this process in mice, we were able to reverse-engineer our findings to accelerate functional maturation in human interneurons.

## Results

Cortical interneurons are generated from a distal source (primarily the MGE) and undergo long-distance migration to their final destination in the neocortex. In the embryonic mouse, this developmental process is rapid: interneurons are generated starting ~e11 and robustly migrate into the cortex by e15 (*Figure 1A*; *Lavdas et al., 1999*; *Marín and Rubenstein, 2001*). As a result, it is tempting to assume that newly born interneurons automatically transition to a migratory state. As a counterexample, however, in human fetal development interneuron migration into the cortex is highly protracted. Previous studies have found that postmitotic interneurons slowly transition to a migratory state to populate the neocortex over the course of many weeks (*Arshad et al., 2016*; *Hansen et al., 2013*; *Ma et al., 2013*). To confirm this, we examined interneurons in fetal cortical sections by immunohistochemistry for Dlx2 (an interneuron marker). Consistent with prior reports, we observed an increase in the density of Dlx2$^+$ interneurons in the cortex over time (15 post-conception weeks [pcw] to 22 pcw) (*Figure 1B*, *Figure 1—figure supplement 1A,B*). We hypothesized that the discrepancy between human and mouse interneuron development may be due to an external cue delivered to the MGE to promote interneuron migration. We therefore examined the developing mouse MGE (from e10.5 to e15.5) by histology and observed a striking increase in vascularization of the MGE and the underlying mantle region during the time period when mouse interneuron migration initiates (*Figure 1C,D*; *Daneman et al., 2009*; *Paredes et al., 2018*). Similarly, we found that vascularization of human fetal MGE slowly increased with developmental age, with robust vascularization not occurring until after 20 pcw (*Figure 1E*, *Figure 1—figure supplement 1C*).

To determine whether there is a causal relationship between MGE vascularization and interneuron migration, we examined *Apcdd1* mutant mice (*Mazzoni et al., 2017*). Apcdd1, a negative regulator of Wnt/β-catenin signaling, is critical for CNS vascular development and blood-brain barrier maturation (*Shimomura et al., 2010*). We examined MGE vessel density and average vessel size in *Apcdd1* GOF (an endothelial cell-specific transgenic gain-of-function) and LOF (whole animal null) mutants at e14.5 and found that, similar to the retina and cerebellum (*Mazzoni et al., 2017*), vascularization in the MGE is decreased in *Apcdd1* GOF and increased in *Apcdd1* LOF mice versus wildtype controls (*Figure 1F–H*). To confirm that a whole animal Apcdd1 null was appropriate for analysis, we performed in situ hybridization for Apcdd1 and found that its expression is confined to vasculature in the embryonic mouse brain (*Figure 1—figure supplement 3*). Importantly, fewer interneurons migrated into the cortex in Apcdd1 GOF mutants, in which vascularization was less extensive compared to wildtype controls. Conversely, a trend toward more calbindin + migrating interneurons were present in *Apcdd1* LOF mutants (p=0.236, paired t-test) in which vessel size and density was greater (*Figure 1I*). These results suggest that the density of endothelial cells within neural tissue is a critical regulator of interneuron migration.

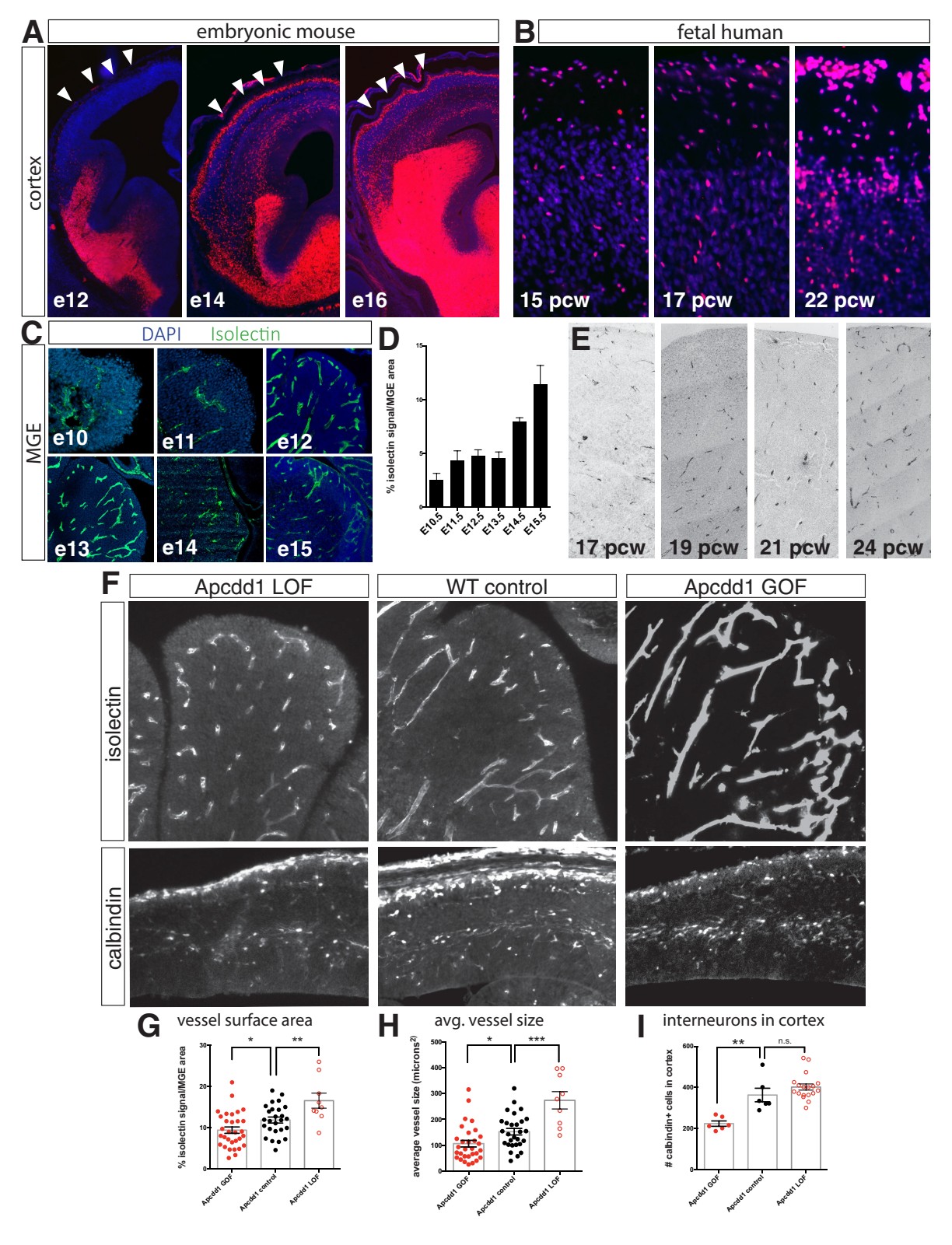

**Figure 1.** Interneuron migration progresses over development and is regulated by medial ganglionic eminence (MGE) vascularization. (A) Coronal sections of Dlx6a^Cre; Ai9 embryonic mouse telencephalon over developmental time. tdTomato + interneurons progressively migrate into the cortex between e12 and e16. Cortex indicated by arrowheads. (B) Dlx2 immunohistochemistry (red) in human fetal cortex shows weeks-long progression of cortical interneurons migrating into cortex. (C) Representative images of isolectin labeling (green). (D) Quantification of isolectin + blood vessel staining

*Figure 1 continued on next page*

*Figure 1 continued*

as a percentage of MGE area for embryos ages e10.5–e15.5. (**E**) CD31 immunohistochemistry in human MGE from various fetal ages. (**F**) Top row, coronal sections of e14.5 MGE. Blood vessels labeled with isolectin in wildtype control, *Apcdd1* loss-of-function and *Apcdd1* gain-of-function mutants. Bottom row, coronal sections of e14.5 embryonic brain labeled with calbindin to show migratory interneurons in isolectin in wildtype control, *Apcdd1* loss-of-function and *Apcdd1* gain-of-function mutants. (**G, H**) Quantification of isolectin + MGE vascularization; (**G**) isolectin + vessel labeling as a percentage of MGE surface area; (**H**) average vessel size in square microns. (**I**) Quantification of total interneurons migrating into cortex/20 μm section. Paired t-test, *p<0.05; **p<0.01; ***p<0.001.

The online version of this article includes the following source data and figure supplement(s) for figure 1:

**Source data 1.** source data for *Figure 1*.
**Figure supplement 1.** Cortical interneurons populate the human fetal cortex over a protracted period.
**Figure supplement 1—source data 1.** source data for *Figure 1—figure supplement 1*.
**Figure supplement 2.** Medial ganglionic eminence (MGE) explants cultured from mouse embryos at various ages.
**Figure supplement 2—source data 1.** source data for *Figure 1—figure supplement 2*.
**Figure supplement 3.** Apcdd1 in situ hybridization signal is confined to endothelial cells in the developing mouse telencephalon.

In order to identify a mechanism by which this occurs, we used an MGE explant culture approach from mouse embryos to quantify the number of migrating neurons at various developmental stages (***Figure 2A***, ***Figure 1—figure supplement 2A–D***). Given that the MGE is progressively vascularized from e10 to e15, we reasoned that MGE explant migration would increase with developmental age.

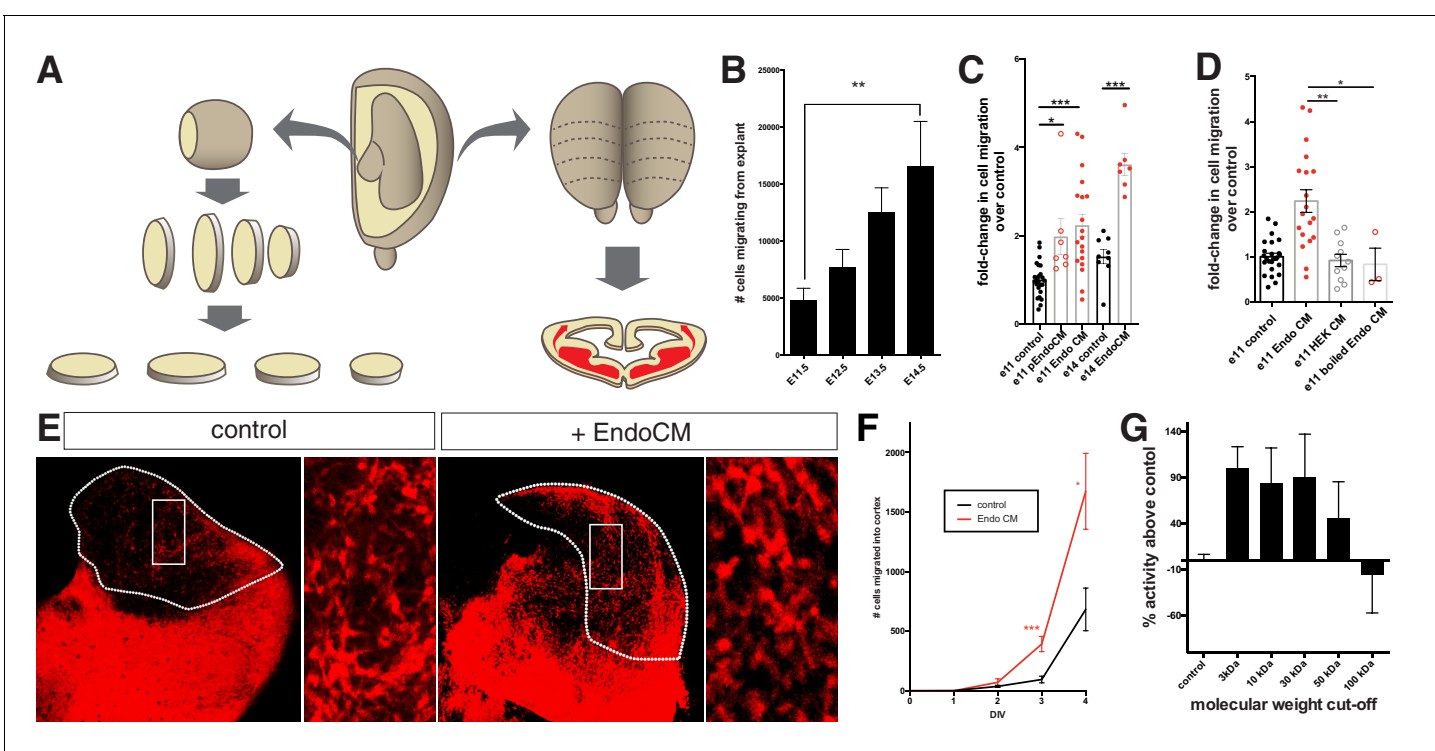

**Figure 2.** Endothelial cell conditioned medium increases interneuron migration in medial ganglionic eminence (MGE) explants and organotypic slice cultures. (**A**) Schematic representation of MGE explant and organotypic slice preparation to assess interneuron migration in vitro. (**B**) Number of interneurons (normalized to MGE surface area) migrating from MGE explants from embryos ages e10.5–e15.5. (**C**) Number of interneurons (normalized to MGE surface area) migrating from e11.5 and e14.5 MGE explants with or without primary culture endothelial cell conditioned medium (p-EndoCM) or immortalized endothelial cell line conditioned medium (EndoCM). (**D**) Number of interneurons (normalized to MGE SA) migrating from e11.5 MGE explants treated with control, EndoCM, HEK 293 conditioned medium (HEK CM) or boiled EndoCM. (**E**) Representative images of DIV4 organotypic slice cultures without (control) or with EndoCM added. Right is higher magnification of boxed region on left. (**F**) Number of Dlx6a^Cre^; Ai9 tdTomato + interneurons migrating into cortex over time in coronally section organotypic slice cultures (DIV 0–4) with or without EndoCM treatment. (**G**) Size fractionation of EndoCM assayed for normalized interneuron migration from e11.5 MGE explants. Paired t-test, *p<0.05; **p<0.01; ***p<0.001.

The online version of this article includes the following source data for figure 2:

**Source data 1.** source data for *Figure 2*.

We further reasoned that in early MGE explants that were not extensively vascularized, that mouse interneurons would be similarly immobile like human interneurons. To normalize for differences in MGE size at different ages, whole MGE tissue was dissected and sectioned at 250 μm. Then, the number of DAPI + migratory cells was normalized to the surface area of the MGE explant. Using this approach, we found a strong linear correlation between the number of migratory cells and MGE explant surface area ($r^2$ = 0.6829) (*Figure 1—figure supplement 2B*). Further, we found that nearly all migratory DAPI + cells were interneurons using the Dlx6a$^{Cre}$ driver line crossed to Ai9 in order to fate-map the migratory lineage (*Monory et al., 2006*; *Figure 1—figure supplement 2C,D*). Classic studies have shown that MGE explants at e14.5 and e15.5 exhibit robust interneuron migration within hours of initial plating (*Bellion et al., 2005*; *Polleux et al., 2002*; *Wichterle et al., 1999*). Consistently, we found that interneurons later timepoint MGE explants exhibited robust migration, whereas explants from earlier timepoints (e10.5–e12.5) had a limited capacity to migrate (*Figure 2B*, *Figure 1—figure supplement 2E*). One possibility is that interneurons possess an intrinsic timer such that, given sufficient time in culture, early timepoint explants would migrate to the same extent as older MGE explants. However, even when cultured for up to a week, early explants did not significantly migrate more after the first 48 hr (data not shown).

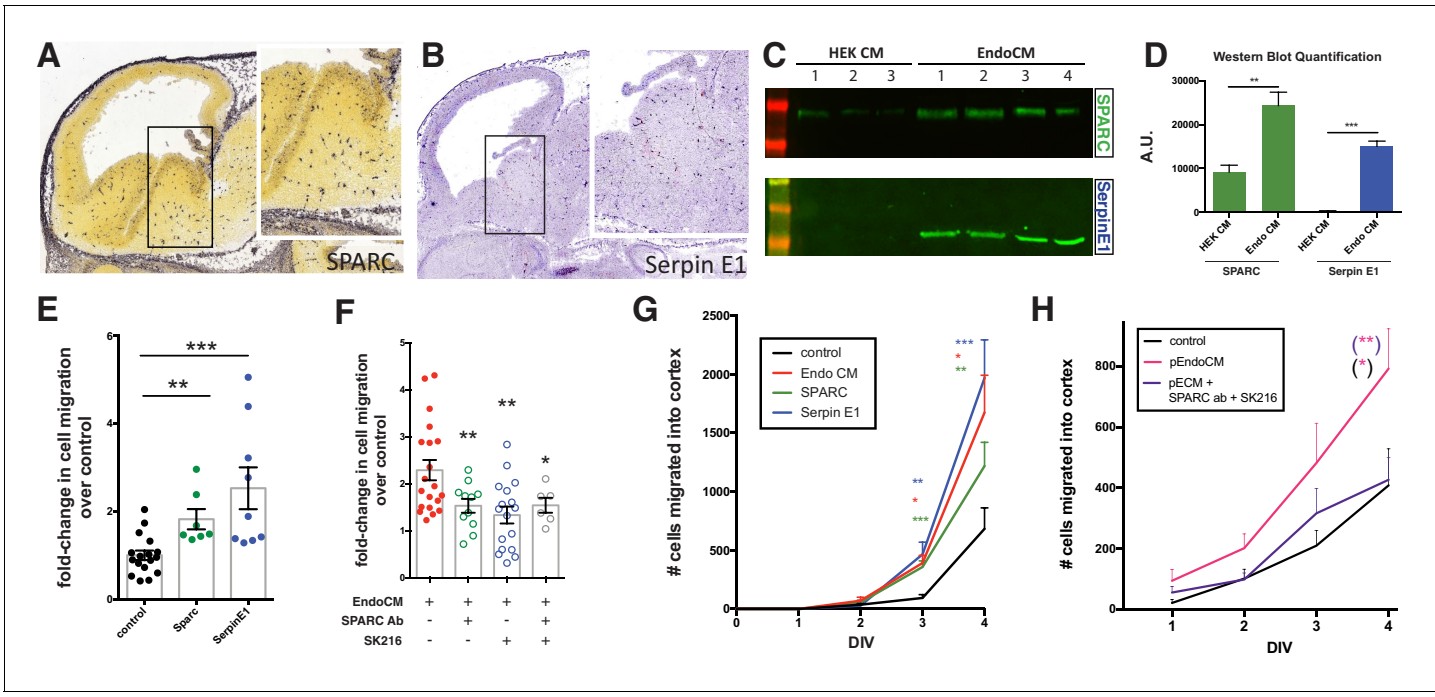

**Figure 3.** Endothelially derived factors, SPARC and SerpinE1, increase interneuron migration and account for most of the biological activity of endothelial cell line conditioned medium (EndoCM). (**A, B**) Parasaggital sections of e13.5 embryonic brain showing in situ hybridization signal for (**A**) SPARC[*] and (**B**) SerpinE1[#]. Insets for (**a**) and (**b**) are higher magnification or boxed regions showing medial ganglionic eminence (MGE) expression. (**C**) Western blots of HEK 293 conditioned medium (HEK CM) (three replicates) and EndoCM (four replicates) for SPARC and SerpinE1. (**D**) Quantification of western band intensity for SPARC (green) and SerpinE1 (blue) in HEK CM and EndoCM. (**E**) Quantification of normalized interneuron migration from MGE explants treated with SPARC (filled green), SerpinE1 (filled blue) compared with control (filled black). (**F**) Quantification of normalized interneuron migration from MGE explants treated with EndoCM (filled red), EndoCM and SPARC function-blocking antibody (unfilled green), EndoCM and SerpinE1 small molecule inhibitor (unfilled blue), and EndoCM with combination of SPARC function-blocking antibody and SerpinE1 small molecule inhibitor (unfilled gray). (**G**) Number of Dlx6a$^{Cre}$; Ai9 tdTomato + interneurons migrating into cortex over time in coronally section organotypic slice cultures (DIV 0–4) with or without EndoCM, SPARC or SerpinE1 treatment. (**H**) Similar organotypic slice culture experiments as in (**G**) with or without primary EndoCM (p-EndoCM), or p-EndoCM depleted with SPARC function-blocking antibody and SerpinE1 small molecule inhibitor, SK216. Paired t-test, *p<0.05; **p<0.01; ***p<0.001. [*] from Allen Brain Atlas (http://developingmouse.brain-map.org/); [#] from GenePaint (http://gp3.mpg.de).

The online version of this article includes the following source data and figure supplement(s) for figure 3:

**Source data 1.** source data for *Figure 3*.

**Figure supplement 1.** RNA-sequencing (RNA-seq) analysis identifies SPARC and SerpinE1 as highly enriched secreted factors in endothelial cell line conditioned medium (EndoCM) versus HEK 293 conditioned medium (HEK CM).

Given the diffuse distribution of blood vessels in the MGE, the most likely mechanism is that endothelial cells produce a paracrine signal to induce interneuron migration. To test this directly, we prepared conditioned medium from primary cultures of embryonic brain ECs (primary culture endothelial cell conditioned medium [p-EndoCM]) and added it to e11.5 MGE explants. Consistent with our hypothesis, addition of p-EndoCM resulted in a robust increase in interneuron migration (*Figure 2C*). We found, however, that primary embryonic brain EC cultures introduced unwanted variability, and for subsequent experiments, we obtained EndoCM from an immortalized human EC line, HBEC5i (*Wassmer et al., 2006*). To further reduce inter-experimental variability, interneuron migration counts were normalized to within-experiment negative (untreated) controls. Hence, subsequent data is presented as fold-change in interneuron migration over controls. Similar to p-EndoCM, conditioned medium from HBEC-5i (EndoCM) also robustly increased e11.5 MGE explant migration (*Figure 2C*). Interestingly, e14.5 MGE migration was also significantly increased by EndoCM (*Figure 2C*). We next tested the effect of EndoCM on interneuron migration in an organotypic slice culture. Here, we used Dlx6a^Cre; Ai9 + e11.5 embryos in order to visualize interneurons as they migrate within a coronal slice of telencephalon from the MGE into the cortex (*Figure 2A,E*). As with MGE explants, EndoCM also significantly increased the rate and overall number of interneurons that migrated into the cortex (*Figure 2F*). As a negative control, we tested the biological activity of conditioned medium from HEK 293 cells (HEK CM). HEK CM did not increase MGE explant migration at either age (*Figure 2D*). Moreover, we found that pre-boiling EndoCM eliminated its biological activity, suggesting a protein source as a regulator of interneuron migration (*Figure 2D*). Finally, we size-fractionated EndoCM and found that biological activity was strongly reduced between 30 and 100 kDa (*Figure 2G*).

In order to identify candidate proteins, we performed bulk RNA sequence analysis on HBEC5i and HEK cells. We screened the dataset for genes with the greatest differential expression, enriched in HBEC5i that produced proteins between 30 and 100 kDa, which were also secreted (Gene Ontology [GO] term: extracellular space) (*Figure 3—figure supplement 1*). After further curation to eliminate membrane-tethered molecules, we obtained a short list of 24 candidates (*Supplementary file 1*). We functionally tested a number of candidates, including VEGF-A (*Barber et al., 2018*) and follistatin; however, two proteins, SPARC and SerpinE1, exhibited the most robust biological activity in their ability to increase e11 MGE migration (*Figure 3*).

We examined the expression of SPARC and SerpinE1 in e14.5 MGE through publicly available expression databases (Allen Brain Atlas and GenePaint) and found that both were specifically expressed in the embryonic CNS vasculature (*Figure 3A,B*). We also confirmed by western blotting that SPARC and SerpinE1 were present in EndoCM at significantly higher levels compared to HEK292 CM (*Figure 3C,D*). We then added recombinant SPARC or SerpinE1 to MGE explants and found that cell migration was increased, in particular with SerpinE1 (*Figure 3E*). We then tested EndoCM in which SPARC and SerpinE1 activity were depleted. We used a function-blocking antibody for SPARC (*Sweetwyne et al., 2004*), whereas SerpinE1 activity was blocked with a small molecule inhibitor (SK216; *Masuda et al., 2013*). These reagents separately significantly reduced the capacity of EndoCM to increase e11.5 MGE migration (*Figure 3F*). However, inhibition of both SPARC and SerpinE1 does not reduce interneuron migration further, suggesting that the two molecules may act on intersecting pathways. Finally, we tested whether SPARC and SerpinE1 could increase interneuron migration in an organotypic slice over time. We found that both proteins significantly increased the rate and overall number of interneurons that migrated into the cortex over time (*Figure 3G*). Finally, we tested whether SPARC and SerpinE1 were active components in p-EndoCM. We found that p-EndoCM increased interneuron migration into the cortex in organotypic slice cultures over control and that the effect was abrogated by adding SPARC function-blocking antibody and SK216 to p-EndoCM (*Figure 3H*).

Previous studies have demonstrated that hSC-interneurons migrate and mature at a slow rate, reminiscent of the protracted time frame of interneuron development in the human fetus (*Arshad et al., 2016*; *Hansen et al., 2013*; *Ma et al., 2013*). Given that SPARC and SerpinE1 elicit interneuron migration in e11.5 MGE explants and organotypic slice cultures, we tested whether these factors might similarly accelerate the developmental time frame for hSC-interneurons. Using a pan-tdTomato expressing human iPSC line (CAG-tdTomato knocked into *AAVS1* locus), we achieved efficient differentiation to ventral telencephalic identity using established protocols (*Bagley et al., 2017*; *Figure 4—figure supplement 1A*). hSC-interneuron differentiation was confirmed using AAV

Dlx5/6-GFP (*Dimidschstein et al., 2016*; *Figure 4—figure supplement 1B*). At day 35 of differentiation (DIFF 35), ventral telencephalic organoids were treated with either SPARC, SerpinE1, or both for 14 days. At DIFF 49, we tested for hSC-interneuron migration by dissociating either untreated or SPARC/SerpinE1-treated organoids. Here, we further divided the groups: one group continued to be exposed to SPARC and SerpinE1 and the other was left untreated (*Figure 4—figure supplement 2A*). We observed a significant increase in migratory distance in the group treated both before and afterward with SPARC and SerpinE1. Importantly, we observed an even greater biological effect when SPARC and Serpin were added to dissociated cells after pre-treatment (*Figure 4—figure supplement 2C*). Thus, in subsequent xenograft experiments, SPARC and SerpinE1 were added as a pre-treatment for 14 days and also added to the cells at the time of transplantation.

Previous studies have xenografted hSC-interneurons into a more functionally relevant setting: neonatal mouse cortex. They found that functional maturation rate is prolonged, requiring ~7 months (*Nicholas et al., 2013*; *Shao et al., 2019*). We then tested the capacity of control and SPARC/SerpinE1-treated ventral organoids to integrate following xenotransplant into immune-compromised (NSG) mouse cortex. We first analyzed transplants 28 days post-engraftment. We confirmed that tdTomato + hSC interneurons were almost all Dlx2+ (*Figure 4A*) and also found that Combo-treated hSC-interneurons migrated significantly further than controls (*Figure 4B,C*). At 56 days post-transplantation, we traced and analyzed the morphologies of hSC-interneurons in 3D and found that they possessed longer processes and branched more extensively (*Figure 4D*, *Figure 4—figure supplement 3*).

Given the improvements in migration and morphology exhibited by treated hSC-interneurons, we tested their functional maturity by employing whole-cell recordings from tdTomato labeled transplanted cells in acute slices of frontal cortex 56 days post-transplant. Consistent with more mature neuronal function, recordings from treatment group neurons showed significantly reduced membrane input resistance, faster membrane time constants, more hyperpolarized action potential (AP) thresholds, and increased ratios of voltage-gated sodium currents versus voltage-gated potassium currents (*Figure 4E–O*). Additionally, other measures of neuronal maturity we tested appeared to collectively trend toward more mature functional phenotypes in the treatment group of neurons, albeit these were not statistically significant (*Figure 4—figure supplement 4*). Specifically, treated cells appear to have larger membrane capacitances (p=0.067), more hyperpolarized resting potentials (p=0.12), larger voltage-gated sodium currents (p=0.20), faster AP rise rates (p=0.21), narrower APs (p=0.22), and generated a greater number of APs (p=0.24) (*Figure 4—figure supplement 4*). Taken together, these electrophysiological profiles suggest that treated neurons may have an accelerated trajectory toward functional maturity.

## Discussion

CNS vascularization is a tightly regulated process that operates in tandem with neuronal and glial cell development (*Paredes et al., 2018*). Here, we demonstrate that not only does vascularization of mouse MGE coincide with interneuron migration, but it also plays a regulatory role since genetic manipulation of vascular pruning affects the number of interneurons migrating into the cortex. Further, two EC-derived paracrine factors SPARC and SerpinE1 induce increased interneuron migration as assessed by in vitro assays. Moreover, ablation of SPARC and SerpinE1 in immortalized or p-EndoCM reduced its capacity increase in interneuron migration. Having demonstrated a critical crosstalk between the neural and vascular compartments in mouse, we leveraged our findings to test if SPARC and SerpinE1 can accelerate hSC-interneuron maturation. We found that hSC-interneurons treated with SPARC and SerpinE1 also show a more robust migration both in vitro and upon xenotransplantation into host mouse cortex. Finally, xenografted hSC-interneurons also exhibit significantly more complex morphologies and more mature electrophysiological properties.

Our findings support previous studies that have linked angiogenesis to pathfinding during interneuron tangential migration (*Barber et al., 2018*; *Li et al., 2018*) and MGE mitosis (*Tan et al., 2016*). Further, radial glia in the MGE undergo a transition during embryonic development where their pial endfeet detach and connect to MGE vasculature (*Tan et al., 2016*). In light of our data, it is tempting to speculate that this rearrangement allows for a more direct communication between ECs and interneurons, thereby facilitating their tangential migration into the cortex. Single-cell RNA-sequencing (RNA-seq) in mouse indicates that SPARC expression is significantly enriched in brain

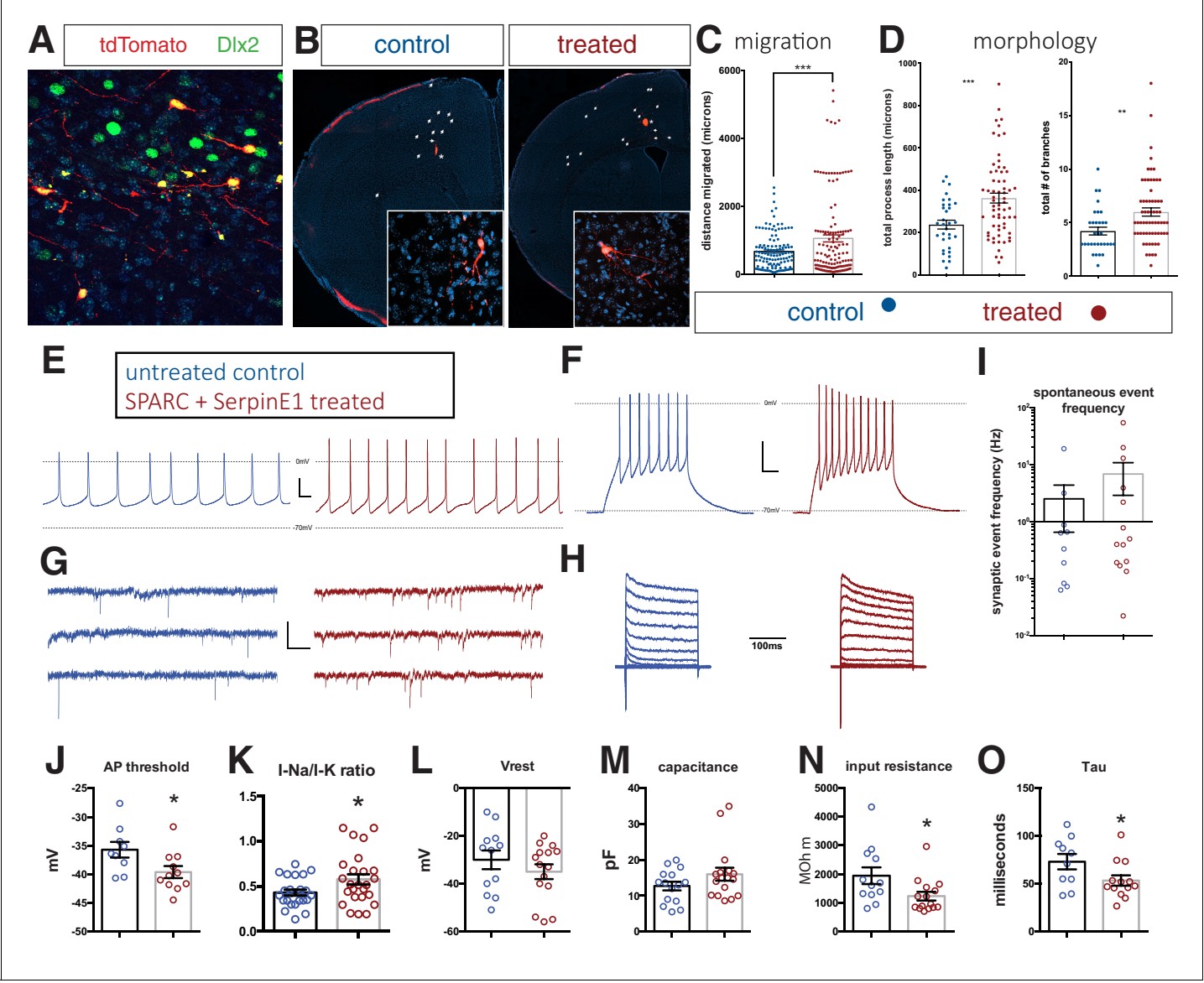

**Figure 4.** Human stem cell-derived interneurons (hSC-interneurons) xenografted into host mouse cortex are more migratory and more morphologically complex with SPARC and SerpinE1 pre-treatment. (A) Immunohistochemistry for Dlx2 (green), Dapi (blue), and tdTomato + hSC interneurons (red) near injection site 1 month post-transplant. (B) Representative images of coronal sections of host mouse cortex 1 month post-transplant of control untreated and SPARC/SerpinE1 pre-treated hSC-interneurons (asterisk denotes transplant site). Arrowheads show tdTomato + hSC interneurons migrating away from transplant site. (C) Quantification of migratory distance of hSC-interneurons from site of injection 1 month post-transplantation. (D) Quantification of hSC-interneuron morphology by average process length (left) and total number of neurite branches (right). (E–O) Whole-cell recordings of hSC-interneurons 2 months post-transplant. Untreated control (blue), SPARC and SerpinE1 pre-treated (red). (E) Representative traces of spontaneous AP firing. (F) Evoked firing upon square pulse depolarization. (G) Representative traces of spontaneous miniature EPSCs. (H) Step recordings of sodium current. (I–O) Quantitative measurements of (I) spontaneous mEPSC frequency (Hz), (J) AP threshold (mV) (p=0.0166), (K) sodium-to-potassium current ratio (p=0.0338), (L) resting membrane potential (p=0.12), (M) capacitance (p=0.067), (N) input resistance (p=0.0155), (O) tau (p=0.0239). Paired t-test, *p<0.05; **p<0.01; ***p<0.001.

The online version of this article includes the following source data and figure supplement(s) for figure 4:

**Source data 1.** source data for *Figure 4*.

**Figure supplement 1.** Human stem cells (hSCs) efficiently differentiate into interneurons.

**Figure supplement 1—source data 1.** source data for *Figure 4—figure supplement 1*.

**Figure supplement 2.** Human stem cell-derived interneurons (hSC-interneurons) migrate further in vitro with SPARC and SerpinE1 treatment.

**Figure supplement 3.** SPARC and SerpinE1-pre-treated human stem cell-derived interneurons (hSC-interneurons) exhibit more complex morphologies after xenotransplantation into host mouse cortex.

*Figure 4 continued on next page*

*Figure 4 continued*

**Figure supplement 3—source data 1.** source data for *Figure 4—figure supplement 3*.

**Figure supplement 4.** Electrophysiological recordings of control and pre-treated human stem cell-derived interneurons (hSC-interneurons) show pre-treated group trends toward greater maturity by most measures.

**Figure supplement 4—source data 1.** source data for *Figure 4—figure supplement 4*.

endothelial cells versus other endothelial populations (log$_2$-fold increase 3.97) (*Hupe et al., 2017*). In the same study, SerpinE1 expression increases in mouse brain endothelial cells, peaking at e14.5, which coincides with the timepoint where we observe robust migration in MGE explants. Both SPARC (*Butler et al., 2016*; *Girard and Springer, 1995*; *Sage et al., 1989*) and SerpinE1 (*Canfield et al., 1989*) are expressed in human endothelial cells.

In humans, interneuron migration during fetal development is a protracted process (*Arshad et al., 2016*; *Hansen et al., 2013*; *Ma et al., 2013*). We found that hSC-interneurons exhibit greater migration and faster functional maturation when treated with vascular factors, SPARC and SerpinE1, suggesting that the maturation of CNS vasculature may similarly regulate the timing of human fetal interneurons migration. During human fetal development, both SPARC and SerpinE1 are expressed at low levels in subpallium at 8 and 9 pcw, and both peak in their expression in the cortex between 37 pcw and 1 year of age (Allen Institute BrainSpan Database, Developmental Transcriptome, https://www.brainspan.org/rnaseq/search/index.html). Further, the MGE (also known as the germinal matrix) is the primary site periventricular hemorrhages in prematurely born babies (*Luo et al., 2019*). It has been hypothesized that this is due to newly formed (and thus weaker) vasculature in the MGE during the third trimester (*Ballabh et al., 2004*; *Ballabh et al., 2007*). As such, it is an appealing possibility that the timing of vascularization, which we posit to be the source of SPARC and SerpinE1 in the MGE, helps to coordinate species-specific aspects of neuronal maturation. Here, delayed vascularization in humans may serve a regulatory role in delaying interneuron migration until pyramidal neurons have achieved sufficient maturity to allow for interneuron circuit integration.

We identified SPARC and SerpinE1 as important proteins that account for most of the EndoCM activity. This is consistent with the described roles for both proteins in other systems. SPARC has been implicated in cell differentiation (*Hrabchak et al., 2008*; *Stary et al., 2005*) and migration (*Arnold and Brekken, 2009*). SPARC is also correlated with increased cancer metastasis and its increased expression in cancer stromal cells results in epithelial-to-mesenchymal transition by downregulation of E-cadherin and upregulation of N-cadherin (*Nagaraju et al., 2014*). SPARC can influence migration through integrins and integrin-linked kinases (*Thomas et al., 2010*) and by modulating the activity of matrix metalloproteases. Indeed integrin signaling also regulates interneuron migration during development (*Stanco et al., 2009*). SerpinE1 (also known as plasminogen activator inhibitor-1) is also implicated in cell migration through the uPA/urokinase pathway (*Grøndahl-Hansen et al., 1993*; *Mahmood et al., 2018*). The urokinase pathway has been linked to interneuron tangential migration by acting through the c-Met receptor and hepatocyte growth factor (*Powell et al., 2001*).

In addition to its effects on intracellular signaling, SPARC and SerpinE1 also modulate interactions between cells and extracellular matrix (ECM). As a matricellular protein, SPARC functions at the interface between cells and ECM. Reduced collagen levels are observed in SPARC null mice and there is compelling evidence that SPARC reduces cell adhesion to ECM, enabling an intermediate state of adhesion that is conducive to migration (*DiMilla et al., 1991*; *Palecek et al., 1997*). SerpinE1 also modulates ECM by binding to vitronectin. This interferes with its binding to integrin αV, resulting in greater integrin αV-fibronectin binding and reduced cell adhesion. In the context of neural stem cells, the balance between integrin αV interactions with vitronectin and fibronectin helps to regulate self-renewal versus differentiation (*Dai et al., 2016*; *Varun et al., 2017*). Vitronectin has a similar role in regulating differentiation in cerebellar granule precursors (*Hashimoto et al., 2016*).

Taken together, we hypothesize that SPARC and SerpinE1 act by tipping the balance toward cell differentiation and migration. SerpinE1 does so possibly by modulating uPA/urokinase signaling, which is active in the MGE (*Powell et al., 2001*) and possibly by interfering with integrin interactions with vitronectin. SPARC likely acts as an anti-adhesive factor that favors cell migration, as has been

described for radially migrating cortical neurons with SPARC-like 1 (*Gongidi et al., 2004*). In future, it will be important to study the molecular basis for how SPARC and SerpinE1 function in regulating interneuron differentiation and migration.

Of note, we empirically determined that human ventral telencephalic organoids require ~14 days pre-treatment with SPARC and SerpinE1 before we could detect enhanced migration in vitro. This may reflect a human-specific difference in responding to endothelial cues compared with mouse interneurons that operate under a more compressed timescale. In fact, this may account for previous observations by Nicholas and colleagues that migration of hSC-interneurons into host mouse cortex occurs at a slow rate over 7 months (*Nicholas et al., 2013*). This may be explained by host vascularization of the xenograft, which has been shown to occur over a period of months (*Mansour et al., 2018*). We hypothesize that pre-treatment with SPARC and SerpinE1 allows our xenograft to bypass host vascularization as a means to trigger migration. Therefore, pre-treatment may prime hSC-interneurons to more rapidly integrate and mature into the host cortex. Of note, we found that levels of SPARC and SerpinE1 were detected a low levels in the mouse MGE at e14.5 (data not shown), consistent with in situ hybridization from Allen Institute and GenePaint (*Figure 3A and B*). This stood in contrast to the higher levels of SPARC and SerpinE1 used to elicit a biological effect in hSC-interneurons. We hypothesize that this is due to context. In culture, SPARC and SerpinE1 are added to the medium and must diffuse through Matrigel-embedded organoids to reach hSC-interneurons. This is in contrast to the embryonic microenvironment in which vasculature is in close contact with interneuron progenitors, and this configuration likely enhances the effects of SPARC and SerpinE1. A number of studies have utilized bioengineering to recapitulate the vascular microenvironment (*Raghavendran et al., 2016*; *Shirure et al., 2017*). Employing SPARC and SerpinE1 in such a context could potentially amplify the effects we demonstrated in this study.

Our findings demonstrate the utility of priming hSC-interneurons for transplantation by pre-inducing them to adopt a migratory state. Although there are likely other impediments to overcome before hSC-interneurons are fully synchronized with the developmental timescale of host mouse cortex, our findings represent an important step toward harnessing the full potential of hSC-interneurons. Cortical interneurons are critical modulators of brain function (*Kepecs and Fishell, 2014*; *Tremblay et al., 2016*) and it is important to develop human interneuron models of disease (*Catterall, 2018*; *Inan et al., 2013*; *Marín, 2012*; *Rapanelli et al., 2017*). Moreover, our pre-treated hSC-interneurons also exhibit more maturity in vitro, suggesting SPARC/SerpinE1 treatment may also be effective when applied to organoid and monolayer approaches. Finally, consistent with previous work (*Karakatsani et al., 2019*), paracrine cues derived from ECs may similarly regulate the development of other neuronal populations. As such, our study may well serve as a general strategy for inducing hSC-neuron functional maturation in other neuronal cell types such as pyramidal neurons.

## Materials and methods

### MGE explant migration assay

Swiss Webster timed plugs were generated to obtain embryos ranging in age from e10.5 to e15.5. MGE was microdissected under sterile conditions into ice-cold Leibovitz's L15 medium (Gibco). MGE tissue was then quickly embedded into low melt agarose (4%) in Leibovitz's L15 medium (Gibco). Embedded blocks were sectioned at 250 µm using a vibratome (Leica VT1000). MGE slices were then individually transferred into four-well slides, then covered with 300 µL Matrigel (Corning) (1:1 dilution) in Neurobasal medium (Gibco). Slides were then transferred to 37°C incubator for 15 min to allow Matrigel to solidify. Then, 300 µL MGE medium containing Neurobasal medium, 2% B27, 1% N2, 1% Glutamax, and 1% penicillin/streptomycin was added on top of Matrigel. After 48 hr, MGE explants were fixed at room temperature in 4% paraformaldehyde/PBS for overnight and DAPI (2 ng/mL) was added to the chamber to label explants. Following one wash with PBS, MGE explants were imaged by confocal microscopy. Confocal images were analyzed using ImageJ plugin, a combination of background removing, cell counting, and maximum intensity in order to automatically segment and quantify DAPI + nuclei.

The following reagents were added to MGE explants diluted in overlying Matrigel:

| Reagent | Working concentration |
|---|---|
| p-EndoCM | 1:1 |
| EndoCM | 1:1 |
| SPARC | 50 ng/mL |
| SerpinE1 | 20 ng/mL |
| SPARC function-blocking antibody (mab 303; *Sweetwyne et al., 2004*) | 1:5 |
| SK216 (*Masuda et al., 2013*) | 100 µM |

For untreated controls, the appropriate vehicle at the same volume was added to x medium. Vehicle for p-EndoCM and EndoCM: Neurobasal; vehicle for SPARC: water; vehicle for SerpinE1: PBS; vehicle for SPARC function-blocking antibody: PBS; vehicle for SK216: water. For controls comparing different treatment groups, vehicle control results were pooled.

## Apcdd1 mouse mutants

Apcdd1 LOF mutants were generated by crossing the *Apcdd1* conditional allele (*Turakainen et al., 2009*) to the heat shock promoter Cre deleter line (*Dietrich et al., 2000*) to generate a whole animal null. The *Apcdd*1 GOF line was generated as described previously (*Mazzoni et al., 2017*) in which a transgenic mouse (TRE3-*Apcdd1*-IRES-*mCherry*) is crossed to *Cadherin-5*::tTa (*Sun et al., 2005*), which results in an endothelial cell-specific Apcdd1 GOF mutant. All mice share the same mixed 129 and C57BL/6J genetic background. We used mixed 129 and C57BL/6J mice as a wildtype control group for comparison.

## Organotypic slice culture migration assay

We employed a modified protocol previously described (*Baffet et al., 2016*). Briefly, Dlx6a$^{Cre}$; Ai9 e11.5 embryos were sac'd and whole brain was isolated by microdissection in aCSF medium bubbled with $O_2$. Brains were embedded in low melt agarose (4%) in aCSF medium and sectioned by vibratome at a thickness of 250 µm. Sections were imaged live daily by confocal microscopy. Images were analyzed using ImageJ plugin trackMate in order to automatically segment and quantify migrating interneurons in cortex.

## Cell lines

Three cell lines were used in this study: two immortalized lines, HEK 293 and HBEC-5i, and one human iPSC line, NCRM1, which was modified by CRISPR to knock-in tdTomato into the AAVS1 locus. All lines were routinely tested for mycoplasma and certified as negative prior to use in experiments.

## Derivation of conditioned medium

Primary mouse BECs were purified from P7 wildtype brains as described (*Daneman et al., 2010*), plated on gelatin-coated T75 flask and grown to confluence in mouse endothelial cell media supplemented with growth factors (VEGF, EGF, FGF-2, insulin) and 10% FBS media (Cell Biologics; Catalog # M1168; Chicago, IL) as described (*Mazzoni et al., 2017*). To collect p-EndoCM, cells were rinsed with PBS and then medium was switched to Neurobasal and 1% Glutamax for 3 days. Supernatant was collected, centrifuged at 4000 × *g* for 15 min, and filtered to remove cells. Then medium was concentrated using a 3000 kDa MWCO Centricon column (Millipore). Concentrated medium was stored at −80℃ until use. Similarly, HBEC-5i cells were maintained in DMEM/F12, 10% FBS, 40 µg/mL endothelial cell growth supplement medium, rinsed with PBS, and then switched to Neurobasal, 1% Glutamax medium for 3 days. EndoCM was similarly spun down, filtrated, and concentrated using 3000 kDa MWCO Centricon column (Millipore). Concentrated EndoCM was also stored at −80℃ until use. For size fractionation experiments, 10, 50, and 100 kDa MWCO Centricon columns (Millipore) were used for media concentration. Control medium was unconditioned medium fractionated with 3000 kDa MWCO column. HEK 293 cells were maintained in DMEM, 10% FBS, 1% Glutamax medium and switched to Neurobasal, 1% Glutamax medium following PBS rinses to collect HEK CM. It was also concentrated using 3000 kDa MWCO Centricon columns.

## RNA sequence analysis

HBEC-5i and HEK 293 cells were switched over to Neurobasal, 1% Glutamax medium in the same manner as if EndoCM and HEK CM were to be collected. Three days later, cells were harvested in Trizol and RNA was purified using directZol miniPrep kit (Zymo). Samples were quality-controlled and sequenced at Novogene using Illumina HigSeq/MiSeq with a sequencing depth of 20 million reads. Sequence data was then analyzed using DAVID to identify high confidence differentially expressed hits of the appropriate molecular weight. These data were further screened by GO search term 'extracellular space' to identify secreted factors. Additional curation was performed to remove membrane-tethered proteins and pseudogenes in order to arrive at a short list of candidates (*Supplementary file 1*).

## Immunohistochemistry and western blots

Immunohistochemistry on *Apcdd1* mutants were as described previously (*McKenzie et al., 2019*). E14.5 Apcdd1 mutant and wildtype control brains were cryosectioned at 16 μm and immunolabeled with Alexa488-tagged isolectin (1:100; Thermo Fisher Scientific); calbindin (1:1000; ImmunoStar). Sections were acquired as tiled maximum projection images for analysis using FIJI for blood vessel quantification and ImageJ plugin cell counting for segmentation and calbindin + cell counts. Western blot antibodies: Goat-anti-SPARC (1:1000; R&DSystems) and rabbit-anti-SerpinE1 (1:1000; Abcam) were used to detect EndoCM and HEK CM concentrated medium, both loaded with 20 μL. Xenograft 50 μm sections were immunolabeled with anti-Dlx2 (1:1000; Millipore).

## Human fetal cortical tissue

We obtained fetal tissue samples for research following induced termination of pregnancy for maternal indications. Sample collection followed the policies of the Columbia University Irving Medical Center Institutional Review Board. IRB waiver AAAS5541 was obtained for non-human subjects research, deemed medical waste.

## Human stem cell differentiation

Parental human iPSC line (NCRM 1, NIH Common Fund Regenerative Medicine Program) was genetically altered by CRISPR-mediated targeting of AAVS1 locus to introduce floxed-stop CAG-boosted tdTomato donor DNA construct. EF1-alpha promotor-driven Cre recombinase was introduced episomally to generate pan-tdTomato + human iPSC line (pan-red line). The pan-red line was differentiated toward ventral telencephalic fate using and adaptation of previously described methods (*Bagley et al., 2017*; *Xiang et al., 2017*). In brief, pan-red hiPSCs were plated into ultra-low attachment u-bottom 96-well plates (9000 cells/well) in neural induction medium containing LDN-193189 (100 mM), SB431542 (10 mM), and XAV939 (10 mM) to form organoids for 10 days. From days 10 to 17, cells changed to neuronal differentiation medium containing N2 and B27 (Invitrogen) with IWP2 (2.5 μM) and SAG (100 nM). From day 18 onward, neuronal differentiation medium also contains BDNF (20 μg/mL), cAMP (125 mM), and ascorbic acid (200 μg/mL). At day 35, SPARC (50 ng/mL) and SerpinE1 (20 ng/mL) are added for 14 days. At day 49, organoids are gently dissociated in EDTA for 5 min, then Acutase 15 min at 37°C for downstream experiments (in vitro migration assays and xenotransplants).

## Xenograft migration and morphology analysis

We performed stereotactic intracranial injections into the right frontal subcortical white matter (coordinates from bregma: 1.0 mm right – 1.0 mm front – 1.0 mm deep) of NRG (NOD.*Cg-Rag1^{tm1Mom}Il2rg^{tm1Wjl}*/SzJ, Jackson Laboratories) mouse pups at P6–9 as previously described, adapted for mouse pups (*Lei et al., 2011*). We injected 12 P5/P6 NRG pups with control and 12 P5/P6 NRG pups with SPARC/SerpinE1-treated cells (~50,000 cells/injection). At 28 and 56 days post-injection, animals were sacrificed and brains were processed for analysis (below). All procedures were performed according to Columbia University IACUC protocol no. AC-AAAV0463.

One month post-transplant, host cortex was sectioned by vibratome at 50 μm. Migratory distance was assessed by scoring tdTomato + cell linear distance from graft site. Two months post-transplant, 300 μm vibratome sections were imaged by confocal microscopy to visualize tdTomato + cell

morphology. Processes were traced using ImageJ plugin NeuronJ to assess neurite length and degree of branching.

## Acute brain slice preparation and electrophysiology

A more complete description of acute brain slice preparation and basic electrophysiology methods has been previously described (*Crabtree et al., 2016*; *Crabtree et al., 2017*).

Briefly, mice were anesthetized with isoflurane, decapitated, and brains were removed quickly and chilled in ice-cold dissection solution, which contained the following (in mM): 195 sucrose, 10 NaCl, 2 $NaH_2PO_4$, 5 KCl, 10 glucose, 25 $NaHCO_3$, 4 $Mg_2SO_4$, and 0.5 $CaCl_2$, and was bubbled with $95\%O_2/5\%CO_2$. Coronal brain slices (~300 μm) centered on the injection site of the transplanted stem cell-derived neurons were cut using a vibratome (a region covering the PFC through the region ~1 mm caudal to the corpus callosum). After slicing, brain slices were immediately transferred to a recovery chamber and incubated at room temperature in recording solution for a minimum of 30 min before recording. Total time between decapitation and procedure end was typically 12–16 min.

At the time of recording, slices were transferred to a submerged recording chamber and continuously perfused with standard aCSF (*Crabtree et al., 2016*). Whole-cell patch-clamp recordings were made using borosilicate glass pipettes (initial resistance, 2.0–5.5 MΩ). An internal solution was used that contained the following (in mM): $KMeSO_4$ 145, HEPES 10, NaCl 10, $CaCl_2$ 1, $MgCl_2$ 1, EGTA 10, Mg-ATP 5, and $Na_2GTP$ 0.5, pH 7.2 with KOH. Solution junction potentials were small and were not corrected.

## Basic electrophysiology

Recordings employed an Axon 700B MultiClamp amplifier, CV-7B headstage, and a Digidata 1440A data acquisition system. All signals were acquired at 10 kHz (100 μs). With the exception of spontaneous synaptic recordings (filtered at 2 kHz), all other signals were filtered at 10 kHz. Cells targeted for recording were identified by red fluorescence. Confirmation of correctly targeted recordings was further validated by observation of a significant reduction in red fluorescent signal in the recorded cell at the end of the recording likely resulting from 'wash-out' of the indicator protein via the recording pipette solution.

### Current clamp recordings

All cells were forced to −70 mV with a small negative current of variable amplitude. Bridge-balance mode was employed to minimize voltage errors and artifacts.

Resting membrane potential was reported as the cell voltage in I = 0 mode observed shortly after whole-cell membrane rupture. The majority of cells were silent at rest leading to an uncomplicated reporting of $V_{rest}$. A minority of cells (typically more hyperpolarized cells), however, displayed spontaneous AP or AP-like events rendering $V_{rest}$ measures somewhat ambiguous. In this subset, $V_{rest}$ was estimated as the midpoint voltage between the AP threshold voltage and deepest hyperpolarization after the AP.

### Membrane time constant

Using small hyperpolarizing current steps, the region of the voltage response from 5 ms after the start of the step to 205 ms within the step was fitted with a single exponential using the standard Clampfit Chebyshev method fitting routine. Accuracy of fits were further confirmed visually.

APs were elicited in cells forced to −70 mV with small (2.5–5 pA), incremental current steps of 500 ms duration. Unless otherwise indicated, reported metrics assessed the first AP elicited from current step recordings.

### AP width

The first AP elicited from current step recordings was used for analysis. AP widths were measured using the standard Clampfit analysis routine 'half-width', the AP width at half-height, and reported as AP width.

### AP threshold

The first AP elicited from current step recordings was used for analysis. The voltage trace of this current step was converted to a time versus dV/dt plot and overlaid onto the original AP voltage trace. AP threshold was then determined visually as the first significant deviation of dV/dt from its baseline rate. All traces analyzed were assessed together in a single analysis session and all traces were displayed at the same time and dV/dt scale to avoid bias in threshold detection. A subset of traces were converted to 'phase-plane' plots (V versus dV/dt) to further validate threshold assignments.

AP maximum rate of rise (dV/dt max) was determined using the standard Clampfit 'maximum rise slope' routine. As with other AP measures, the first AP elicited from current step recordings was used for analysis.

### Voltage clamp recordings

All cells were held at $-70$ mV unless otherwise noted. As cell resistances were high (typically ~1–2 G$\Omega$), pipette series resistances were low (typically <10 M$\Omega$), and maximal elicited currents were relatively small (typically 1–3 nA), series resistance compensation was not employed. The cell capacitance reported is that reported by the 700B amplifier which is the fast component which represents contributions from the soma and proximal process compartments.

Voltage-activated currents were elicited with incremental 100 ms voltage steps (from $-100$ mV to $+50$ mV) of 100 ms duration in cells held at $-70$ mV. Membrane resistance was determined from the current response of the voltage step to $-80$ mV ($\mu$V, $-10$ mV). Transient inward sodium currents were reported as the maximal sodium current elicited (I-Na$_{max}$), typically observed at the step to $-20$ or $-30$ mV. Outward potassium currents were reported as the maximal potassium current elicited (I-K$_{max}$) at the step to $+50$ mV. The I-Na/I-K ratio was then derived from these two values.

Spontaneous synaptic transmission was recorded from cells held at $-70$ mV in the absence of TTX. A standard recording duration of 3 min was employed. The observed synaptic events are likely dominated by glutamatergic synaptic events as the reversal potential for GABA-A currents of our solution combination was ~$-60$ mV. The frequency of synaptic events was highly variable between cells with some cells 'silent' (or nearly so) while other cells had synaptic event frequencies in excess of 20 Hz. Due to the extreme variability in these synaptic event profiles, herein we present only exemplary traces of the synaptic activity we observed.

### Statistics

Electrophysiological parameters were compared with pairwise t-tests between conditions. t-Tests were one-tailed with a directional hypothesis of 'more maturity' of the metric in the treated group.

## Acknowledgements

We would like to thank members of the Au lab for their edits and careful reading of the manuscript, especially Melissa McKenzie and Luke Nunnelly. Thanks to Meer Patel for much of the preliminary groundwork that enabled the quantification of MGE explant migration assay. Many thanks to Dr. Rolff Brekken (UT Southwestern) for his kind gift of SPARC function-blocking antibodies. Finally, special thanks to Dr. Barbara Corneo and Alejandro Garcia Diaz (Columbia Stem Cell Initiative Stem Core Facility) for their kind gift of a pan-tdTomato expressing NCRM-1 human iPSC line.

## Additional information

### Funding

| Funder | Grant reference number | Author |
| --- | --- | --- |
| Whitehall Foundation | 2016-12-137 | Edmund Au |
| Irma T. Hirschl Trust | | Edmund Au |
| National Institutes of Health | R03MH119443-01 | Edmund Au |
| National Institutes of Health | R01NS117695 | Edmund Au |

The funders had no role in study design, data collection and interpretation, or the decision to submit the work for publication.

## Author contributions

Matthieu Genestine, Data curation, Formal analysis, Investigation, Writing - original draft, Project administration; Daisy Ambriz, Saptarshi Biswas, Data curation, Investigation; Gregg W Crabtree, Data curation, Formal analysis, Investigation; Patrick Dummer, Mouse husbandry, Assisted with experiments in Figure 3H; Anna Molotkova, Michael Quintero, Investigation; Angeliki Mela, Resources, Methodology; Huijuan Feng, Joseph A Gogos, Data curation, Formal analysis, Supervision; Chaolin Zhang, Resources, Data curation, Supervision; Peter Canoll, Supervision; Gunnar Hargus, Resources, Supervision; Dritan Agalliu, Edmund Au, Conceptualization, Data curation, Formal analysis, Supervision, Funding acquisition, Methodology, Writing - original draft, Writing - review and editing

## Author ORCIDs

Chaolin Zhang [iD] http://orcid.org/0000-0002-8310-7537
Edmund Au [iD] https://orcid.org/0000-0003-3190-9711

## Ethics

Human subjects: We obtained fetal tissue samples for research following induced termination of pregnancy for maternal indications. Sample collection followed the policies of the Columbia University Irving Medical Center Institutional Review Board. IRB waiver AAAS5541 was obtained for non-human subjects research, deemed medical waste.

Animal experimentation: This study was performed in strict accordance with the recommendations in the Guide for the Care and Use of Laboratory Animals of the National Institutes of Health. All of the animals were handled according to approved institutional animal care and use committee (IACUC) protocol (AC-AAAZ6451) of Columbia University.

## Decision letter and Author response

Decision letter https://doi.org/10.7554/eLife.56063.sa1
Author response https://doi.org/10.7554/eLife.56063.sa2

# Additional files

## Supplementary files

• Supplementary file 1. Top differentially expressed genes between HEK 293 and HBEC-5i. HEK 293 and HBEC-5i cell lines were pre-treated in serum-free medium as if they were being used to collect conditioned medium. Cells were lysed and RNA was collected for bulk RNA-sequencing (RNA-seq) analysis. Top differentially expressed genes were screened by protein size and using the Gene Ontology (GO) term extracellular space in order to identify candidates listed in this table.

• Transparent reporting form

## Data availability

Bulk RNA-seq datasets for HEK293 and HBEC 5i have been uploaded to NCBI GEO under Accession Number: GSE146991.

The following dataset was generated:

| Author(s) | Year | Dataset title | Dataset URL | Database and Identifier |
|---|---|---|---|---|
| Au E, Genestine M | 2021 | bulk RNA-seq analysis comparing HEK293 and HBEC 5i cell lines | https://www.ncbi.nlm.nih.gov/geo/query/acc.cgi?acc=GSE146991 | NCBI Gene Expression Omnibus, GSE146991 |

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
