## [Decision Letter]

**Acceptance summary:**

The authors demonstrate the important role for vascular derived factors in the migration and maturation of sub cortically derived interneurons. The reviewers were enthusiastic about the work but had some criticisms that the authors have addressed fully in their revised version.

**Decision letter after peer review:**

Thank you for submitting your article "Vascular-Derived SPARC and SerpinE1 Regulate Interneuron Tangential Migration and Maturation" for consideration by *eLife*. Your article has been reviewed by 3 peer reviewers, and the evaluation has been overseen by a Reviewing Editor and Jonathan Cooper as the Senior Editor. The reviewers have opted to remain anonymous.

The reviewers have discussed the reviews with one another and the Reviewing Editor has drafted this decision to help you prepare a revised submission.

As the editors have judged that your manuscript is of interest, but as described below that additional experiments are required before it is published, we would like to draw your attention to changes in our revision policy that we have made in response to COVID-19 (https://elifesciences.org/articles/57162). First, because many researchers have temporarily lost access to the labs, we will give authors as much time as they need to submit revised manuscripts. We are also offering, if you choose, to post the manuscript to bioRxiv (if it is not already there) along with this decision letter and a formal designation that the manuscript is 'in revision at *eLife*'. Please let us know if you would like to pursue this option. (If your work is more suitable for medRxiv, you will need to post the preprint yourself, as the mechanisms for us to do so are still in development.)

Summary:

In this manuscript, Genestine et al. demonstrate that two endothelial cell-derived paracrine factors, SPARC and SerpinE1, enhance interneuron migration in mice and may facilitate rapid migration and maturation of human stem cell derived interneurons. Authors characterize embryonic vascularization of the median ganglionic eminence (MGE) and its effect on interneuron migration into the cortex. Following the observation that an increase in MGE vascularization closely coincides with this migration event, they demonstrate that endothelial cell conditioned media enhances interneuron motility in vitro. The authors then identify two secreted protein factors in the endothelial-conditioned media – SPARC and Serpin E1 – necessary and sufficient for increased interneuron migration in their in vitro system. In a subsequent series of experiments, the authors show that treatment of iPSC-derived human interneurons with these factors significantly enhances the cells' maturation and motility in a mouse xenograft. While the significance of CNS vasculature in regulating interneuron migration and maturation in mice is well established, the main advance provided by this study is the potential role of SPARC and SerpinE1 in accelerating human interneuron developmental maturation. In general, this study and its conclusions are supported by a complementary array of interneuron migration assays (in mice and transplanted hSC-INs) and whole cell electrophysiological recording of interneurons. But from a neurodevelopmental perspective the manuscript's central claim – that vascular-derived SPARC and SerpinE1 regulate interneuron migration – remains insufficiently supported. While the experiments presented hint at the existence of an exciting endothelial-neuronal interaction it falls short of establishing its operation in vivo.

Essential revisions:

1. Loss-of-function in vivo studies would be necessary to confidently establish a role for these secreted proteins. While this study provides some very interesting data, the authors have not shown that it is specifically vascular endothelial-derived SPARC and SerpinE1 that regulates interneuron migration and maturation. It is widely appreciated that primary brain endothelial cells rapidly de-differentiate and lose key aspects of their organotypic character when cultured in vitro. As a result, the generalizability of the SPARC and Serpin E1 phenotypes to an in vivo developmental setting cannot be assumed. The authors did not specifically block or delete the function of vascular-derived SPARC and SerpinE1. The authors test the requirement of vascular-derived SPARC and SerpinE1 by inhibiting the ability of endothelial conditioned media to induce migration of interneurons in explants. Unfortunately this experiment does not specifically block vascular-derived SPARC and SerpinE1 as there is evidence that SPARC is expressed by other cell types in the brain (astrocytes Kucukderily et al. 2011, 10.1073/pnas.1104977108, and figure 3A VZ/SVZ and 3B cortical VZ/SVZ). To show the requirement of endothelial SPARC and SerpinE1, the authors could do any of the following experiments: (A) generate conditioned media from mouse primary endothelial cells, deplete CM of SPARC, SerpinE1 or both, and compare if depleted CM induce less migration than intact CM in brain slices. (B) Test if conditional deletion of SPARC or SerpinE1 in endothelial cells prevents or reduces interneuron migration. There are floxed allele mice for SPARC (Ramu 2019, 10.1016/j.ebiom.2019.09.024) and SerpinE1 (Jiang 2017, doi.org/10.1111/acel.12643). (C) Instead strengthening the human work, so that in the end this work can at least advance our ability to accelerate human neuron differentiation process.

2. A clear indication that endothelial cells is the source of the physiological levels of SPARC is needed. SPARC has been studied extensively in astrocytes which could be the source. What is the developmental pattern of expression of SPARC and SerpinE1 in human embryonic development? Do they correspond to the slow rate of interneuronal migration and maturation in human cortex. Publicly available database (e.g. Allen Brain) might have this information, or authors could potentially provide some simple experiments to address this.

3. Is the amount of SPARC and SerpinE1 used in the treatment of hSC-Ins at physiologically relevant levels (either in the context of mouse or human cortex)?

4. The authors do not attempt to explore a mechanism-of-action for the endothelial-derived cues they identify. Which receptors do they interface with to interact with interneurons? Broadly, what intracellular changes do they induce to promote a migratory state? At the very least, discussions of potential answers to these questions are needed for the work to represent a significant scientific advance.

---

## [Author Response]

Essential revisions:1. Loss-of-function in vivo studies would be necessary to confidently establish a role for these secreted proteins. While this study provides some very interesting data, the authors have not shown that it is specifically vascular endothelial-derived SPARC and SerpinE1 that regulates interneuron migration and maturation. It is widely appreciated that primary brain endothelial cells rapidly de-differentiate and lose key aspects of their organotypic character when cultured in vitro. As a result, the generalizability of the SPARC and Serpin E1 phenotypes to an in vivo developmental setting cannot be assumed. The authors did not specifically block or delete the function of vascular-derived SPARC and SerpinE1. The authors test the requirement of vascular-derived SPARC and SerpinE1 by inhibiting the ability of endothelial conditioned media to induce migration of interneurons in explants. Unfortunately this experiment does not specifically block vascular-derived SPARC and SerpinE1 as there is evidence that SPARC is expressed by other cell types in the brain (astrocytes Kucukderily et al. 2011, 10.1073/pnas.1104977108, and figure 3A VZ/SVZ and 3B cortical VZ/SVZ). To show the requirement of endothelial SPARC and SerpinE1, the authors could do any of the following experiments: (A) generate conditioned media from mouse primary endothelial cells, deplete CM of SPARC, SerpinE1 or both, and compare if depleted CM induce less migration than intact CM in brain slices. (B) Test if conditional deletion of SPARC or SerpinE1 in endothelial cells prevents or reduces interneuron migration. There are floxed allele mice for SPARC (Ramu 2019, 10.1016/j.ebiom.2019.09.024) and SerpinE1 (Jiang 2017, doi.org/10.1111/acel.12643). (C) Instead strengthening the human work, so that in the end this work can at least advance our ability to accelerate human neuron differentiation process.

We have performed organotypic slice culture migration assays using primary culture endothelial conditioned medium with or without SPARC function-blocking antibody and SK216 Serpin E1 inhibitor (Figure 3H). Here, we found that pEndoCM significant increased the number of interneurons migrating into the cortex and that the biological activity of pEndoCM was attenuated in the presence of function-blocking ab and SK216. We have updated the figure legend, results and discussion to reflect these new data.

2. A clear indication that endothelial cells is the source of the physiological levels of SPARC is needed. SPARC has been studied extensively in astrocytes which could be the source. What is the developmental pattern of expression of SPARC and SerpinE1 in human embryonic development? Do they correspond to the slow rate of interneuronal migration and maturation in human cortex. Publicly available database (e.g. Allen Brain) might have this information, or authors could potentially provide some simple experiments to address this.

We observe robust migration from e14.5 mouse MGE, at which point we assert the effects of SPARC and SerpinE1 has already exerted their effects. This timepoint precedes most of astrogliogenesis and therefore astrocytes are unlikely to be the source for the process we are studying, at least in the case of mice. In human fetal development, astrocytes could be the source since astrocytes (or at least astrocyte progenitors) are detected at 19 pcw (Holst et al., 2019). Although, notably, in this same study, the authors not that the MGE has far fewer astrocyte progenitors than neighboring LGE and CGE.

As for expression of SPARC and SerpinE in human fetal brain, we found evidence in support of increased expression of both SPARC and SerpinE1 late in gestation. From the Allen Institute developmental transcriptomic database (BrainSpan), there is low level expression of SPARC and SerpinE1 in the subpallium at 8 and 9 pcw. In the fetal cortex, expression for both peaks at 37 pcw through to 1 year of age. Further, previous reports find expression of SPARC (Butler et al., 2016; Girard and Springer, 1995; Sage et al., 1989) and SerpinE1 (Canfield et al., 1989) in human endothelial cells. Single cell RNA-seq in mouse indicates that SPARC expression is significantly enriched in brain endothelial cells vs. other endothelial populations (log2-fold increase 3.97) (Hupe et al., 2017). In the same study, SerpinE1 expression increases in mouse brain endothelial cells, peaking at e14.5, which coincides with the timepoint where we observe robust migration in MGE explants. Expression data outlined above has been incorporated into Discussion:

“Our findings support previous studies that have linked angiogenesis to pathfinding during interneuron tangential migration (Barber et al., 2018; Li et al., 2018) and MGE mitosis (Tan et al., 2016). […] In the same study, SerpinE1 expression increases in mouse brain endothelial cells, peaking at e14.5, which coincides with the timepoint where we observe robust migration in MGE explants. Both SPARC (Butler et al., 2016; Girard and Springer, 1995; Sage et al., 1989) and SerpinE1 (Canfield et al., 1989) in human endothelial cells.”3. Is the amount of SPARC and SerpinE1 used in the treatment of hSC-Ins at physiologically relevant levels (either in the context of mouse or human cortex)?To try and get a sense of how much SPARC and SerpinE1 are expressed in mouse MGE at e14.5, we analyzed by western blot and found low, but detectable levels. However, much more SPARC and Serpin E1 was detected in EndoCM, pEndoCM and also in when the lane was loaded with the amount of SPARC and SerpinE1 added to hSC-IN organoids.These results are not particularly surprising. For one vascular cells are a small fraction of the total cells in e14.5 MGE tissue and it would therefore be expected to be detected at low levels in relation to total tissue. Further, it is difficult to calculate the amount of SPARC and SerpinE1 made biologically available in the microenvironment where endothelial cells are signaling immediately adjacent to interneuron progenitors. It is reasonable to assume, however, that a lower level would be needed. This would be especially true of a matricellular protein such as SPARC, which would be interacting with cells in ECM-rich microdomains. Indeed, recent work demonstrates that artificially recapitulating a microdomain environment can greatly enhance the overall effect of a ligand (Raghavendran et al., 2016; Shirure et al., 2017).In contrast, we added SPARC and Serpin E1 for 14 days into the medium surrounding Matrigel-embedded organoids. In this context, much more SPARC and SerpinE1 would be needed to diffuse through the Matrigel and into the organoids. That said, we thank the reviewer for raising this important point. We have added a passage to the Discussion raising the possibility of recapitulating the vascular/MGE microenvironment for added efficacy in future studies:“Of note, we found that levels of SPARC and SerpinE1 were detected a low levels in the mouse MGE at e14.5 (data not shown), consistent with in situ hybridization from Allen Institute and GenePaint (Figures 3A and B). […] A number of studies have utilized bioengineering to recapitulate the vascular microenvironment (Raghavendran et al., 2016; Shirure et al., 2017). Employing SPARC and SerpinE1 in such a context could potentially amplify the effects we demonstrated in this study.”4. The authors do not attempt to explore a mechanism-of-action for the endothelial-derived cues they identify. Which receptors do they interface with to interact with interneurons? Broadly, what intracellular changes do they induce to promote a migratory state? At the very least, discussions of potential answers to these questions are needed for the work to represent a significant scientific advance.

We replaced the paragraph in the Discussion with 3 new paragraphs that addresses potential mechanisms of action for both SPARC and Serpin E1:

“We identified SPARC and SerpinE1 as important proteins that account for most of the activity in EndoCM. […] In future, it will be important to study the molecular basis for how SPARC and SerpinE1 function in regulating interneuron differentiation and migration.”